# Primate Inferotemporal Cortex Neurons Generalize Better to Novel Image Distributions Than Corresponding Deep Neural Networks Units

**Marliawaty I Gusti Bagus**[1,2,3,*]  **Tiago Marques**[1,4,5,6,*]  **Sachi Sanghavi**[1]

**James J. DiCarlo**[1,4,5,†]  **Martin Schrimpf**[1,4,5,†]

[1]McGovern Institute for Brain Research, MIT
[2]Center for Digital Technology and Management, LMU
[3]Graduate School of Systemic Neurosciences, LMU
[4]Department of Brain and Cognitive Sciences, MIT
[5]Quest for Intelligence, MIT
[6]Champalimaud Clinical Centre, Champalimaud Foundation
[*/†] Joint first/senior authors

## Abstract

Humans are successfully able to recognize objects in a variety of image domains. Today's artificial neural networks (ANNs), on the other hand, struggle to recognize objects in many image domains, especially those different from the training domain. It is currently unclear which parts of the ANNs could be improved in order to close this generalization gap. Previous work postulates that a linear readout from 500-800 sites in high-level visual cortex (IT) predicts human behavioral patterns in core object recognition tasks [1, 2]. While internal representations of some ANNs are partially aligned to primate IT firing rates [3], their behavioral outputs when generalizing outside the training distribution are strongly misaligned [4]. In this work, we compared the out-of-domain generalization power of neural populations in primate IT with corresponding neural populations in ANNs. Specifically, we fit a linear decoder on image representations from one domain and test transfer performance on twelve held-out domains, evaluating primate IT representations vs. representations in ANN penultimate layers. To compare fairly, we scale the number of each ANN's units so that its in-domain performance matches that of the sampled primate IT population (i.e. 73% binary-choice accuracy from 71 IT neural sites). We find that the sampled primate population achieves, on average, 68% performance on the held-out-domains. Comparably sampled populations from ANN model units generalize less well, maintaining on average 60%. This is independent of the number of sampled units: models' out-of-domain accuracies consistently lag behind primate IT. These results suggest that making ANN model units more like primate IT will improve the generalization performance of ANNs.

## 1 Introduction

Ample behavioral evidence shows that humans are able to recognize objects in a wide variety of image domains better than current computational models [5, 6, 4], indicating that yet-to-be-created ANN models (i.e. those more like the human brain) could generalize far better than current models.

4th Workshop on Shared Visual Representations in Human and Machine Visual Intelligence (SVRHM) at the Neural Information Processing Systems (NeurIPS) conference 2022. New Orleans.

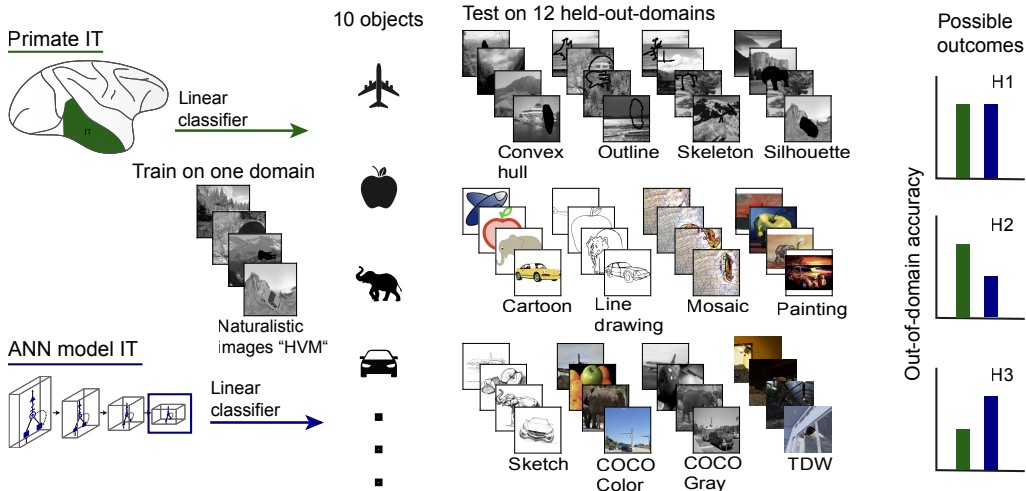

Figure 1: **Comparing the generalization performance of primate IT vs. ANN model representations with linear decoding.** We constructed image sets with 2400 images from 13 different domains, including e.g. naturalistic, cartoon, and sketch images. All domains contain objects from the same 10 categories, but in different image styles. To evaluate the goodness of representations (primate IT recordings, or ANN penultimate layers), we trained linear classifiers on the representations corresponding to one domain (naturalistic images, "HVM" [1]) and, without any further training, tested their ability to classify images from 12 held-out domains into 10 categories. We can thereby directly compare primate vs. model representations while keeping the readout fixed in order to determine whether model IT representations are as good as primate IT (H1), worse (H2), or better (H3).

However it remains unclear which parts of the current ANN models are most in need of improvement, and what could be fixed to create new improved models.

The standard scientific conceptual framework of how humans and non-human primates recognize objects is that, based on the retinal activity evoked by each image, the visual ventral stream produces a powerful feature set via hierarchical transformations from early visual cortex V1 through inferotemporal cortex (IT), and that features of the IT neural population are then linearly combined via downstream neural circuits to produce a class choice behavioral decision [7, 1, 2]. The processing from the image up to the IT population activity is often referred to as the image "encoder" and the readout of IT features into the behavior is referred to as the "decoder". Accordingly, ANN models that are mapped onto this framework can be separated into encoder (e.g. features at their penultimate layer) and decoder (their linear transformation into class labels). These models assume the same kind of decoder as the standard neuroscience model (multi-class linear classifiers) and we will take advantage of that model-to-brain alignment assumption here.

In this work, we asked whether primate IT and model representations lead to different generalization performances when fitting a linear decoder on these representations from one image domain and then evaluating it in 12 unseen domains (Figure 1). Specifically, we use neural recordings from 71 primate IT sites in response to images from 13 different domains, and compare them to a "matched" number of penultimate units in several neural network models. We make 3 novel contributions:

1. We evaluate the generalization performance of a linear decoder fit on primate IT recordings (71 sites) and find that it achieves 73% binary-choice accuracy in-domain (naturalistic images) and 68% on average across 12 held-out domains (e.g. cartoons and sketches).

2. We determine a model-specific scaling factor that lets us compare model units to primate sites, based on the number of model units necessary to reach the performance of a fixed number of primate sites within an image domain.

3. Comparing the generalization performance of populations of primate IT sites vs. a matched population of units in ANN models, we find primate IT to outperform current deep neural

networks, particularly for mosaic, outline, and silhouette image domains (model average performance is 60%).

## 2 Results

We first constructed an image dataset consisting of images categorized into 13 different domains: synthetic naturalistic images ("HVM"; generated by randomly pasting 3D objects onto random naturalistic backgrounds [1]), 4 domains obtained by converting HVM images: silhouette, convex-hull, skeleton, and outline [5], cartoons, line drawings, mosaics, paintings, sketches [5], photographs from Microsoft COCO [8] in color and grayscale, and renders from ThreeDWorld (TDW) [9] (see Figure 1 for examples). Each domain contains at least 60 images, with at least 6 images for each of the 10 object classes (apple, bear, bird, car, chair, dog, elephant, person, plane and zebra).

### 2.1 Primate IT sites generalize to unseen image distributions

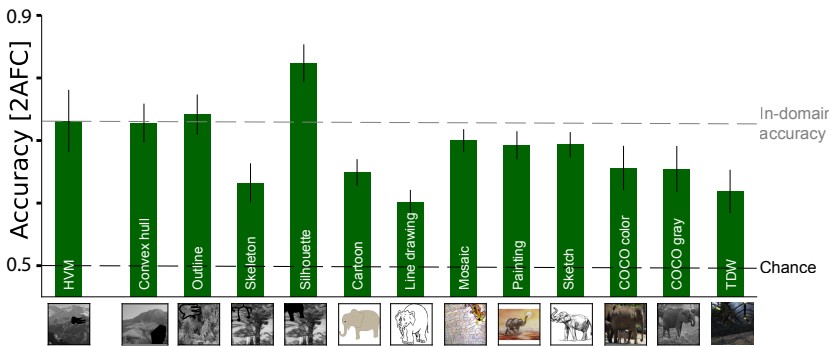

Figure 2: **Generalization performance of primate IT representations (71 sites) with a linear decoder.** We fit a linear classifier to primate IT representations (71 recording sites) corresponding to images from one domain (naturalistic images, HVM), and tested its performance on representations corresponding to held-out HVM images ("in-domain, I.D.") and images from 12 held-out domains ("out-of-domain"). Bars represent 2 alternative-forced-choice accuracy (chance = 50%) on each domain, and error bars are the standard deviation over train and test image sampling. Performance for each domain is well above chance, on average 68% over held-out domains (73% in-domain).

We implanted a macaque monkey with three Utah micro-electrode arrays in inferior temporal cortex (IT; posterior, central, and anterior, $4.4mm \times 4.2mm$ span with 96 electrodes per array). All surgical and animal procedures were performed in accordance with National Institutes of Health guidelines and the Massachusetts Institute of Technology Committee on Animal Care. We pooled all IT sites and used only sites with consistent responses across image repetitions (Appendix A).

To evaluate the utility of IT representations for transferring to new domains, we fit a linear classifier to predict object labels. Specifically, we trained a ridge classifier on the IT population responses to 100 HVM images and evaluated it on 20 held-out HVM images and 50 images for each held-out-domain, using the class with the higher probability in each two-alternative-forced-choice (2AFC) trial (Appendix B).

Within-domain (HVM; testing on held-out images), this classifier achieves a binary-choice performance of 73% with 71 IT sites (Figure 2). Testing the same classifier (without further training) on images from the 12 held-out domains, the linearly decoded primate IT population on average achieves 68% accuracy and performs well above chance for all domains. The most difficult domains for the sampled IT sites were line drawing (60%), TDW (62%), and skeleton (63%). These same domains had also been found to be behaviorally difficult for humans to recognize [5]. This is consistent with the standard neuroscience hypothesis that IT linear decodes underlie primate behavioral choice [1], and provided further confidence that our planned comparison of the IT population with various ANN neural populations would be meaningful.

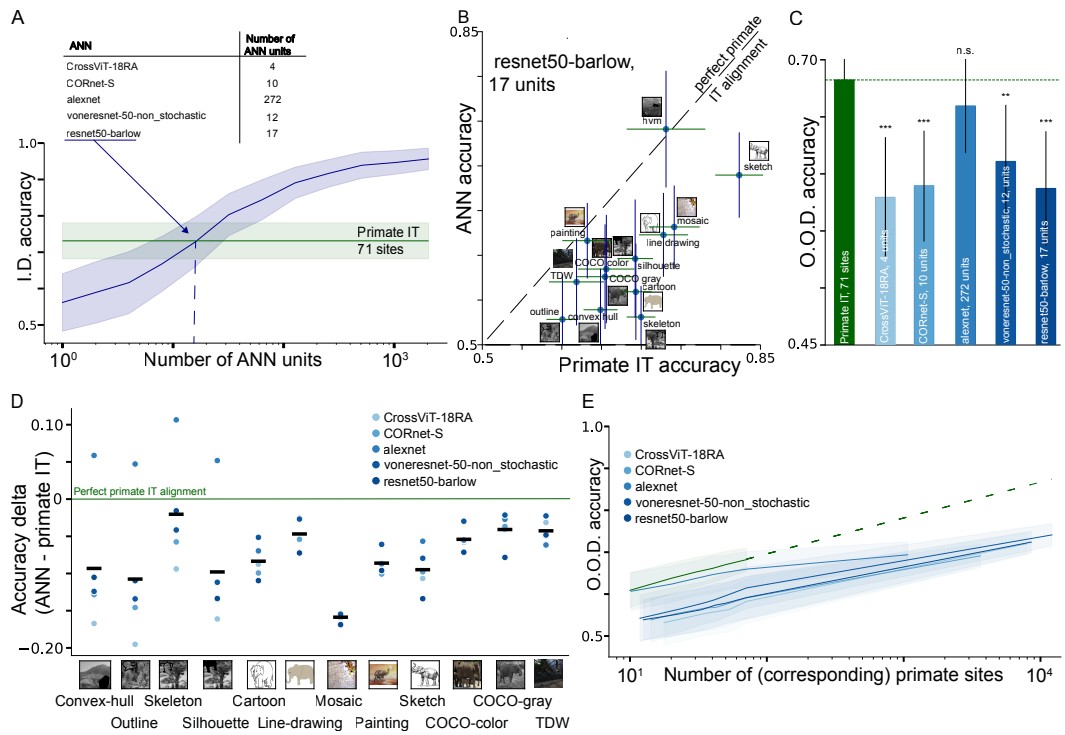

Figure 3: **ANN model unit populations do not generalize as well as comparable primate IT neural populations. A** Scaling model units to primate sites: To keep the number of ANN model units comparable to the 71 primate IT sites, we determined how many units of each ANN are required to reach the same in-domain (I.D.) performance as with the primate IT representations. Sampling more units from the ANNs (x-axis, log scale) improves performance and eventually reaches the performance of 71 primate IT sites. The table specifies for each ANN model how many units are necessary to reach this intercept. Plot is showing one model, resnet50-barlow, see Figure B.1 for all models. **B** Alignment of ANN and primate IT generalization performance: We fit a linear decoder on the representations of ANN units (sub-sampled from the penultimate layer according to the scaling factor determined in *A*) in response to HVM images and evaluated its 2AFC accuracy on held-out HVM images and the 12 held-out domains. Primate IT accuracy as in Figure 2. Error bars are standard deviation over train and test image samples (and unit samples for the ANN). Showing resnet50-barlow, see Figure B.2 for all models. **C** Out-of-domain (O.O.D.) accuracy of primate IT representations compared to corresponding ANN unit representations: Averaging over the performances in 12 held-out domains (see *B*), we computed mean O.O.D. performances for each model. Error bars are standard deviation over domains. Mean O.O.D. performance for each ANN was tested against primate IT representation with a two-sample t-test, testing the null-hypotheses that both samples have the same expected averaged value. **D** Performance difference between ANN models and primate IT per domain: For each ANN and each domain, we subtract the performance of primate IT from ANN accuracy; i.e. deltas below zero signify that the ANN performs worse than primate IT. Horizontal green line is 0, i.e. no accuracy difference. **E** ANN models lag behind primate IT even when using more sites: We plot the averaged out-of-domain accuracy of each ANN when using up to all the units in the penultimate layers (log x-axis, number of units scaled to corresponding primate sites following *A*). For primate IT, we logarithmically extrapolate up to 10.000 sites. Shaded error bars are standard deviation over the 12 held-out domains. Gap between primate IT's and ANNs' O.O.D. performance is consistent when using different numbers of primate IT sites to determine ANNs' scaling factors (see Figure B.3).

## 2.2 ANN models generalize worse than primate IT

We next evaluated how the representations of current ANN models held up as a basis set for transfer performance compared to primate IT representations. We chose models based on the highest scores on

Brain-Score [3, 10] IT benchmarks [1, 11, 12] and tested *CrossViT-18RA*, a dual-stream transformer with rotational invariance and adversarial training [13], *VOneResnet50*, where the first layer is a V1-like frontend model of the primate primary visual cortex [14], *resnet50-barlow*, a Resnet trained with self-supervised redundancy reduction [15], *CORnet-S*, a shallow recurrent network [16], and *alexnet* [17] as a standard baseline despite poorer Brain-Score performance.

To keep model units comparable to the limited number of primate IT recordings (71 sites), we estimated a scaling factor of how many units per model correspond to a primate site. To do this, we computed each model's in-domain performance (using the same linear classifier as for primate IT, Section 2.1) when sampling different numbers of sites from the penultimate layer. We then determined the number of model units needed to achieve identical performance to the primate IT population (i.e. 71 IT recording sites). We found that the corresponding number of units varies between models, e.g. 17 in resnet50-barlow and 4 in CrossViT-18RA (Figure 3A and B.1). Alexnet required an order of magnitude more units (272) than other models to reach 73% in-domain performance. For the following analyses, we randomly sampled units from each model up to the number determined above.

Next, we evaluated the capabilities of each of these unit-matched model sub-populations in the same way we had evaluated the IT population: using a linear classifier trained on in-domain images and then tested on each set of out-of-domains (O.O.D.) images. Comparing each ANN averaged O.O.D. performance against primate IT (two-sided-two-sample t-test, testing if sample averages are different from one another), we found that most models did not generalize as well as primate IT (Figure 3C), supporting H2 from Figure 1. For four out of five models the average O.O.D. performance was significantly different from averaged primate IT performance. Only alexnet showed no significant difference to primate IT but also requires an order of magnitude more units to match 71 IT sites' in-domain performance than the other ANNs. To further examine the performance difference between primate IT and ANN representations on the single held-out-domain level, we did a one-way Friedman Chi-square test, testing if mean differences of held-out-domains are different from one another. We found that models show a significantly worse performance for mosaic Figure 3D; p<10e-9).

### 2.3 Performance gap between primate IT and ANNs is consistent when varying numbers of sites and units

Since our analyses so far all operate on 71 primate IT sites (and the corresponding number of model units), we tested the generalization power of neural populations when varying the number of IT sites and corresponding ANN model units. While fewer IT sites decreases accuracy on held-out domains, the accuracy of models with a corresponding number of units (determined by the scaling factor determined in Section 2.2) equally decreases, such that the gap between primate IT and models is consistent (Figure 3E).

In an exploratory analysis, we further compared the full ANN models (i.e. using all units in the penultimate layer) to the performance of larger primate IT neural populations (Figure 3E green dashed line). Even when using all units of each model's penultimate layer, generalization accuracies trail below the primate IT performance (extrapolated via logarithmic $y = a \times log(x) + b$). For instance, VOneResnet-50 with all 2048 units achieves 75% average accuracy on the 12 held-out domains whereas we extrapolate primate IT with the corresponding number of 12,117 sites to achieve 87%. This generalization gap is independent of the number of primate IT sites that were used to determine ANNs' scaling factor (Figure B.2).

## 3 Discussion

Combining primate neural recordings, state-of-the-art artificial neural network (ANN) models, and linear decoding, we found that primate IT neural populations generalize better than corresponding ANN model unit populations. While linear classification from 71 primate IT neural sites maintains 93% accuracy on held-out domains relative to in-domain performance on average, ANNs maintain only 82% relative performance on average across models. ANNs fall most behind primate IT for mosaic, outline, and silhouette images. These results suggest that making ANN models at their "IT" level more like primate IT should help close the behavioral generalization gap of ANNs observed in previous literature [5, 6, 4].

**Data limitations.** Our analyses were limited by a neural dataset of 71 primate IT recording sites, and a relatively small image count such that we were only able to train on 100 images. Since the

number of training images was consistent across all systems under study (primate IT and ANNs), we do not expect this factor to influence relative performance differences. Dealing with the inherent sampling limitation for primate recordings, we estimated how many ANN model units correspond to the 71 recording sites, but it is possible that these scaling factors do not perfectly reflect the unit-to-site correspondence. We thus also compared the full number of model units to an extrapolated number of primate IT sites, which showed a consistent gap between primate and ANN performance. More neural recordings are necessary to validate extrapolation predictions.

**Linear decoding.** Following previous linking assumptions [1, 2], we used a linear classifier to transform neural representations into classification choices. Since we used the same linear decoding for primate IT and ANN populations, we can directly compare representations and find ANNs to lag behind primate IT generalization performance. This result suggests the "encoder" part of ANNs (transformations up to penultimate layer) as a culprit of non-human-like generalization performance but does not absolve the involvement of the linear decoder, i.e. the decoder might still require fixing. A strong test would be to evaluate whether primate IT populations with a linear decoder perfectly generalize to new domains and fully predict human behavioral choices in all situations.

**Making models more primate IT-like.** Our results suggest that making ANN representations more like primate IT will improve generalization performance. How can we align ANN representations to inferotemporal cortex? Further improvements on ImageNet performance [18] no longer seem to improve neural alignment [10]. An alternative might be to optimize for neural recordings directly [e.g. 19, 20]. We predict such improvements on aligning models to the brain to yield concrete payoffs in transfer benchmarks, highlighting a synergistic link between brain science and machine learning with respect to generalization performance.

## Acknowledgments and Disclosure of Funding

This work was supported by the Netzwerk Engagement e.V. (M.IGB.), the Lothar and Sigrid Rohde Foundation (M.IGB.), the PhRMA Foundation Postdoctoral Fellowship in Informatics (T.M.), the Semiconductor Research Corporation (SRC) and DARPA (J.J.D., M.S.), Office of Naval Research grant MURI-114407 (J.J.D.), the Simons Foundation grant SCGB-542965 (J.J.D.), the MIT-IBM Watson AI Lab grant W1771646 (J.J.D.), the Takeda Fellowship in AI and Health (M.S.), and the Friends of the McGovern Fellowship (M.S.).

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

# Supplementary Material

## A    Primate recordings

**Passive fixation task:**    We presented images on an LCD monitor ($1920 \times 1080$ pixels, 60 Hz refresh rate; ASUS VS248H-P) placed in front of the animal. The images were presented on a grey background, in the central $8°$ of the animal's visual field. Standard rapid serial visual presentation (RSVP) protocol was used, with 100 ms on and 100 ms off durations, a 300 ms pretrial fixation period, and 6–10 images per trial. The animal was head-restrained, and trained to passively fixate on a central white dot ($0.2°$). Real-time eye-tracking (SR Research, EyeLink II) was employed to ensure that eye jitter did not exceed $\pm 2°$, otherwise the trial was aborted and data discarded. For successful trials, the animal was rewarded with juice. Each image was presented pseudo- randomly once in an experimental block. The blocks were repeated multiple times, resulting in 44–61 repeat recordings of each testing image. Stimulus display and reward control were managed using the MWorks Software (http://mworks-project.org).

**Electrophysiological recodings:**    During each recording session, neural activity was amplified ($1\times$ gain) and digitized (sampled at 20 kHz) using the the Intan RHD Recording Controller (Intan Technologies, LLC). The raw voltage signals were bandpass filtered offline using a second-order Elliptic filter (300 Hz to 6 kHz, 0.1 dB passband ripple, 50 dB stopband attenuation), before being thresholded to obtain the muti-unit spike counts. The threshold was set for each recording session and each recording site individually, at 3 standard deviations above the noise. To ensure accurate stimulus locking, the spikes were aligned to a photodiode trigger (Thorlabs, DET36A2) attached to the display screen.

Only IT sites that were deemed "stable" across recording days were retained. To pick these sites, we showed each image in multiple repetitions, and computed the similarity of one half of image repetitions (averaged) to the other half of repetitions of the same images (also averaged). In detail, we measured the internal consistency, defined as the Spearman-Brown corrected split-half reliability, of responses for a fixed set of 25 out-of-set naturalistic images (termed the normalizer set) shown at the beginning of every recording session ($r > 0.75$), and for HvM images shown in a separate experiment ($r > 0.7$). Of 288 recording sites (three arrays $\times$ 96 electrodes), we retained 71 sites across anterior, central, and posterior inferior temporal cortex.

The multi-unit response to an image was taken as the average spiking rate in the 70–170ms temporal window after stimulus onset. The spiking rates were normalized for each site (and session) by subtracting its mean response to the 25 normalizer images and then dividing by the standard deviation of its response over those normalizer images. This was done to compensate for day-to-day variations and allow meaningful comparisons across days and sites.

## B    Decoder training

In each of the 100 splits, we randomly drew 100 training and 20 held-out-images from HVM (in-domain) as well as 50 testing images from the 12 held-out-domains (Figure 1). Per split, we trained a linear ridge classifier on the HVM training images and predicted the probabilities for each class for the in-domain held-out images and each of the 12 held-out-domain images. The classifier's 13 class probabilities were further converted into a single choice for each 2AFC target-distractor trial by choosing the class with the higher probability. Accuracies were averaged (mean) over all 100 splits. Alpha and intercept hyperparameters were fitted via leave-one-out cross-validation.

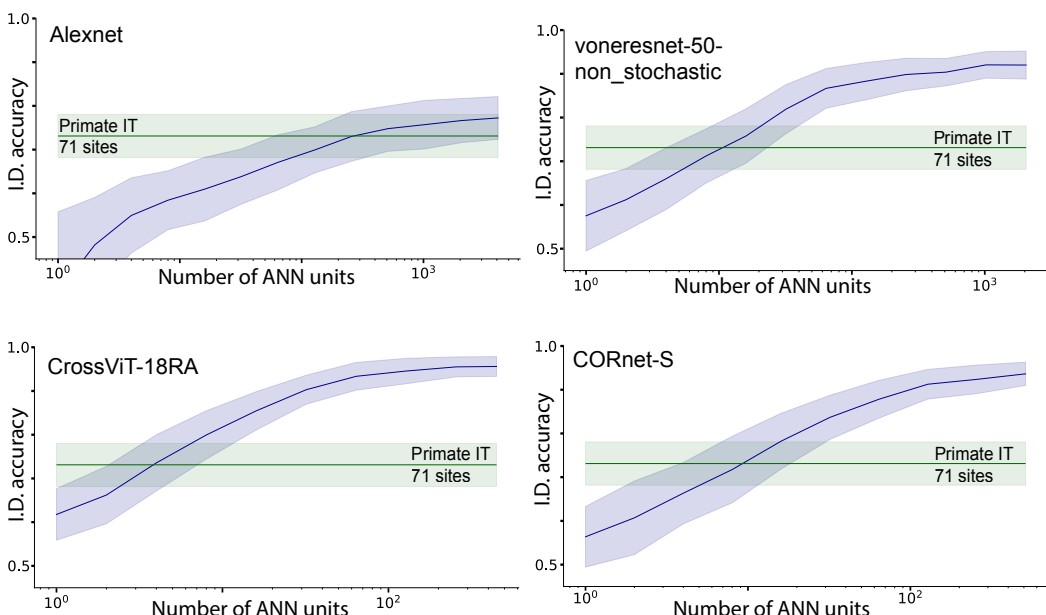

Figure B.1: **Scaling model units to primate sites.** As in Figure 3A, for the remaining ANN models.

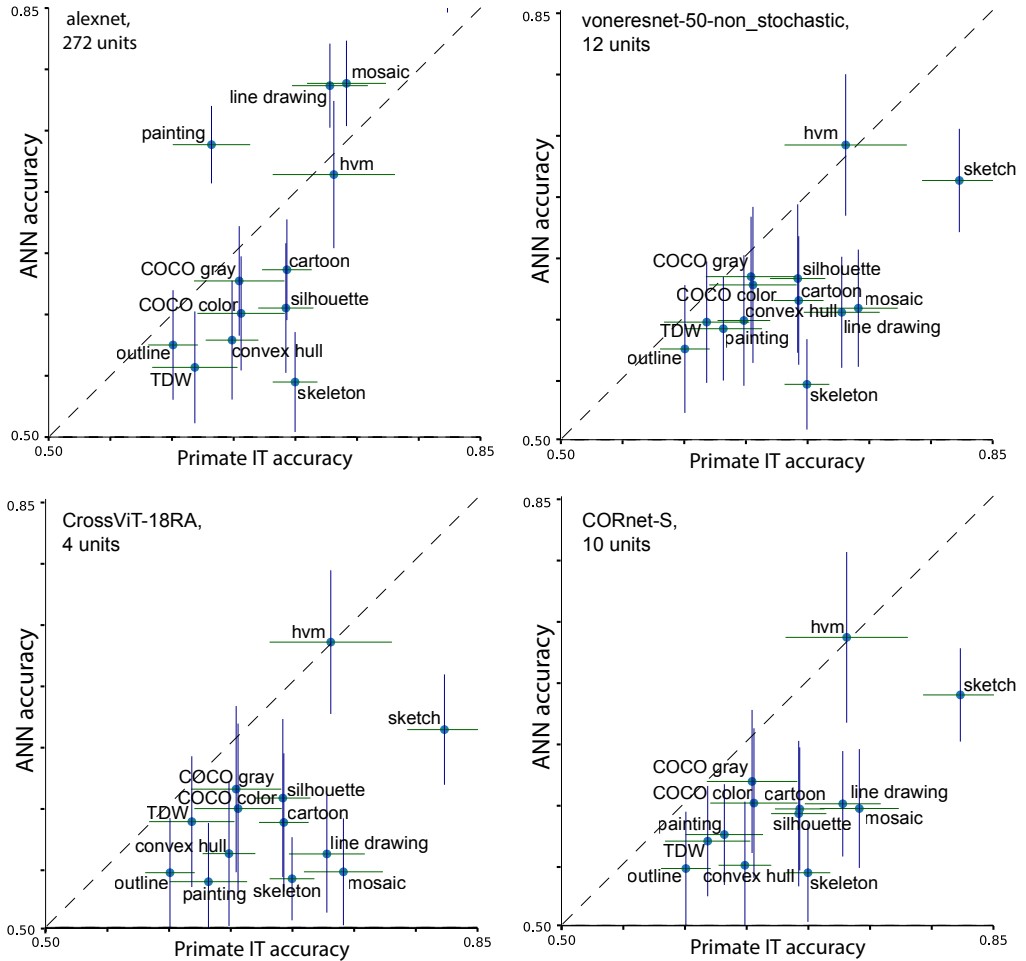

Figure B.2: **Alignment of ANN and primate IT generalization performance.** As in Figure 3B, for the remaining ANN models.

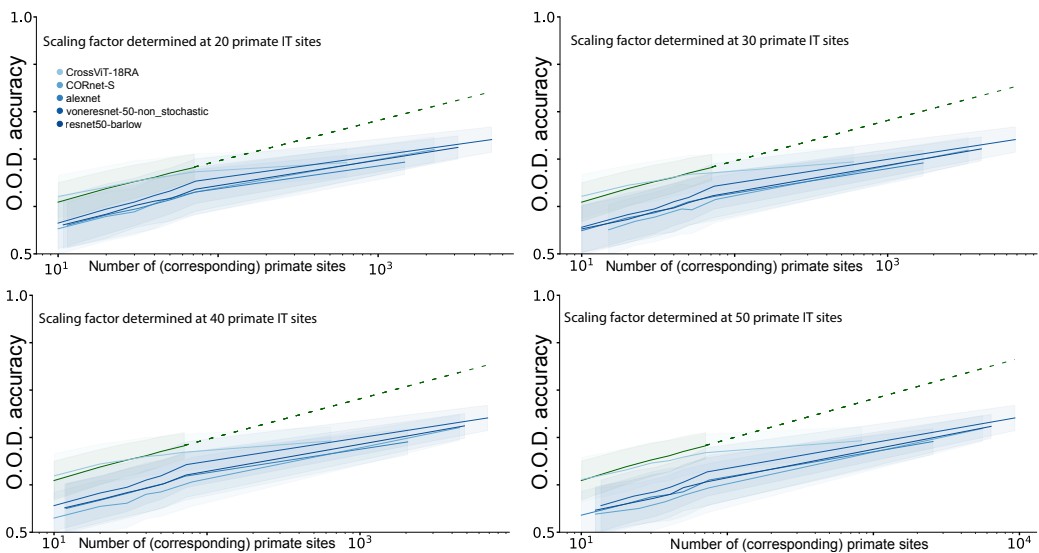

Figure B.3: **Generalization gap between ANNs and primate IT for different unit-sites-matches**
As in Figure 3E, for different numbers of primate IT sites that were used to determine ANNs' scaling factor.

