# OpenReview forum: "Primate Inferotemporal Cortex Neurons Generalize Better to Novel Image Distributions Than Analogous Deep Neural Networks Units"
_NeurIPS.cc/2022/Workshop/SVRHM — SVRHM Poster_

### Official Review · Reviewer_6oCf · 2022-10-11
**This work compared the ability of IT neurons in a macaque to ANNs on a test of generalization.  This paper is a solid contribution that incorporates experimental work and simulation, and is well placed for the SVRHM workshop.**

**Rating:** 8
**Confidence:** 4

**Review:**

This paper compares the ability of ANNs to IT neurons within the macaque on a test of style generalization.   The authors constructed a new data set consisting of 13 styles for each of 10 categories.  The monkeys and networks were pre-trained on natural images and the ability of their neurons to classify images from the other 12 styles were compared.  The general findings are that when in-domain performance was equated by varying the number of neurons, the ANNs were worse at classifying the out of domain images.  Also, even when all of the neurons in the penultimate layer of the ANNs were used for classification, they still could not reach IT performance.

The key contribution of this work is to compare generalization performance between biological and artificial systems, which is of theoretical importance because it is precisely the inability to function well outside of a training regime that highlights the relative insufficiency of ANNs compared to biological visual systems.  The research is well conducted with reasonable efforts to benchmark performance, and also to extrapolate beyond the available data.  Moreover the creation of a dataset with 12 non-natural styles is a useful tool for understanding the specific shortcomings of ANNs with regard to generalization.

The methods of comparing biological and ANNs is  commonly used but the results in this case are novel by comparing generalization ability when in-domain performance is closely matched. The implications for better understanding the specific gap between ANNs and  biological vision. The results are pushed farther by documenting which particular kinds of images have the poorest performance and as an extreme example using all available neurons in the penultimate layer of the comparison models.

In the general discussion it would be good to emphasize that this is “relative” accuracy so that a casual reader is not misled.  The writing on the whole is very good and will be accessible to the audience of the workshop.  The topic of the article is also very well pitched for the SVRHM workshop.

It has been noted before that smaller networks generalize better and that is true here with the ancient alexnet besting the other ANNs in relative accuracy.  This should be emphasized. Another analysis that would be helpful is to compare generalization performance against the # of parameters in each of the candidate models.

It is well known that ANNs tend to classify based on texture rather than shape. It would seem that a very  important comparison would be with the model of Geirhos et al 2015  https://doi.org/10.48550/arXiv.1811.12231 which provides a variety of training enhancements to enhance the ability of ANNs to classify based on shape rather than texture.

---

### Official Review · Reviewer_SFpW · 2022-10-13
**interesting yet preliminary results**

**Rating:** 6
**Confidence:** 3

**Review:**

This is an interesting study. The authors compared the generalization performance of an IT neural population and “neural populations” from several ANN models.

The results are preliminary but interesting.

This reviewer has substantial concerns about whether the comparisons are indeed fair, and how sampling might have affected these results.

The training/validation/test routine could be described more clearly. What’s the decision rule used for the images from the held-out domains?

The definition of “domain” is slightly confusing.

---

### Official Review · Reviewer_uhrS · 2022-10-14
**A really neat paper with interesting findings**

**Rating:** 8
**Confidence:** 4

**Review:**

This paper explores and compares the ability of deep neural networks and primate IT to generalise to out of distribution settings. This is acheived by training linear classifiers on a sample of primate IT neurons (and a corresponding sample of penultimate layer neurons in the DNNs) and evaluating performance on a collection of data domains.

**Pros**
- The idea of matching the number of units sampled from the DNNs to acheive the same performance as the IT recordings is convincing as a fair way of comparing the two settings.
- The choice to compare to models which obtain a high BrainScore adds another layer to this work as a useful differentiator for brain-machine similarity assessments.
- The paper is very thorough and transparent regarding its experimental approach.

**Cons**
- I feel that this work would benefit from a detailed analysis which frames these findings in the context of the wider deep learning literature. For example, I would be greatly interested to read an assessment of how these results relate to the work on shape-bias as it seems like the data domains that DNNs perform less well on, such as line drawings, could perhaps be explained by a lack of shape-bias.

Overall, this is a really neat paper that is a joy to read, feels fair in its analyses, and advances the state of comparisons between DNNs and biology.

---

### Official Review · Reviewer_F78w · 2022-10-15
**Good paper, interesting analysis, needs some details.**

**Rating:** 7
**Confidence:** 4

**Review:**

The authors explore the lower generalization capability of ANN to out-of-training distribution stimuli compared to IT neurons.  Specifically they are interested in which part of ANNs, the encoder or decoder, is responsible. The authors present a novel way of fairly comparing a limited set of neurons in the IT cortex and units in the penultimate layer of ANNs. This is done by finding the number of ANN units that are needed  to achieve the same performance using a linear decoder that is trained on the IT responses. Next, they evaluate the performance of the selected set of units on out-of-distribution images and compare it to the performance of the IT neurons. By extrapolating out-of-distribution performance for different numbers of IT neurons and comparing the out-of-distribution performance of an equivalent number of units in several different ANNs, they show that IT neurons maintain superior performance. This is true even when all the units in the penultimate layer of an ANN are used.

The paper is generally well written and the proposed method  of comparing IT neurons and ANN units is very interesting. However, I think that some of the conclusions are too strong. The authors claim the encoder part of ANNs (representation in the penultimate layer) is primarily responsible and making ANN responses more like IT will lead to better generalization. Ideally, including some results in this direction would help support this conclusion.  Moreover, the actual decoder of ANNs for classification tasks typically uses a non-linear softmax function.  It may not be ideal to replace it with a linear decoder to evaluate its capability. Some discussion on this may be useful.

I think the paper is  good and worth accepting. Further work will make it very compelling.

Below I have included small changes that can improve the current submission:

[1] Many abbreviations are missing their full expanded forms:
  (a) Line 43 HVM
  (b) Line 55 TDW
  (c) Line 69  2AFC

[2]  Line 53  The description of the different image domains can be expanded. Maybe in the Appendix if space is limited.

[3] Line 71. How is achieving a binary choice performance of 71%  consistent with ~800 neurons are required to match human categorization performance.

[4 ] The conversion from the original multi-classification task to the 2AFC task is not explained.

[5] For IT cortex, the words sites and neurons are used interchangeably ? Are they the same ?

[6] It is hard to tell the different ANNs apart in Figure 3D and 3E. Consider using different colors or markers.

[7] Figure 3 Caption. What is 10.000 neurons.?